# Assessing Area under the Curve as an Alternative to Latent Growth Curve Modeling for Repeated Measures Zero-Inflated Poisson Data: A Simulation Study

Daniel Rodriguez 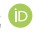

Department of Urban Public Health and Nutrition, School of Nursing, La Salle University,
1900 West Olney, Avenue, Philadelphia, PA 19141, USA; rodriguezd@lasalle.edu

**Abstract:** Researchers interested in the assessment of substance use trajectories, and predictors of change, have several data analysis options. These include, among others, generalized estimating equations and latent growth curve modeling. One difficulty in the assessment of substance use, however, is the nature of the variables studied. Although counting instances of use (e.g., the number of cigarettes smoked per day) would seem to be the best option, such data present difficulties in that the distribution of these variables is not likely normal. Count variables often follow a Poisson distribution, and when dealing with substance use in the general population, there is a preponderance of zeros (representing not using). As such, substance use counts may approximate a zero-inflated Poisson distribution. Unfortunately, analyses with zero-inflated Poisson random variables are not easily accommodated in many types of software and may be beyond access to most researchers. As such, an easier method would benefit researchers interested in assessing substance use change. The purpose of this study is to assess the area under the curve as an option when dealing with repeated measures data and contrast it to one popular method of longitudinal data analysis, latent growth curve modeling. Using a Monte Carlo simulation study with varying sample sizes, we found that the area under the curve performed well with different sample sizes and compared favorably to the performance of latent growth curve modeling, particularly when dealing with smaller sample sizes. The area under the curve may be a simpler alternative for researchers, especially when dealing with smaller sample sizes.

**Keywords:** latent growth curve modeling; area under the curve; Monte Carlo simulation study; substance use; zero-inflated Poisson distribution; longitudinal data



## 1. Introduction

Researchers have a variety of options to capture substance use (e.g., smoking combustible cigarettes or vaping) when having repeated measures (longitudinal data). Popular choices include ordinal (i.e., increasing levels of use) or binary (uses versus does not use) discrete variables. While using ordinal and binary discrete variables is simple for readers to conceptualize, their use for researchers interested in assessing behavior change requires understanding more complicated data analysis methods with varying assumptions and, in some cases, having specialized software. Two popular approaches to conducting repeated measures analysis are generalized estimating equations (GEE) and linear mixed-effects models. GEE models repeated measures variables with various distributions, including normal and negative binomial random variables, when data are missing completely at random (MCAR), although the lack of a likelihood function precludes model comparisons. [1–3] Researchers specify the link function and the correlation structure to obtain population parameter estimates. Extensions of GEE can include missing data that are not MCAR [2,4]. Linear mixed-effects models account for the correlation among clusters of data, such as within person (repeated measures in which the data are clustered within the person), and

the assessment of fixed and random effects [5]. Although GEE, linear mixed-effects models, and other extensions of generalized linear models can be conducted in most conventional statistical computer packages (e.g., SPSS and SAS), other popular options such as latent growth curve modeling (LGCM) require specialized software, especially when the data are discrete. Further, discrete binary variables may mask finer changes in substance use over time, such as when progressing from smoking weekly to daily. Ordinal variables, while providing a finer differentiation of use over time than binary variables, can involve a large number of response patterns, with the number of response patterns increasing with each additional level and repeated measure [6,7].

Perhaps the most natural way to assess substance use is to count the number of occurrences of use per unit time (e.g., combustible cigarettes smoked per day, the number of times vaping electronic cigarettes per week, or the number of drinks of alcohol per month). Such data are also discrete but are likely to follow a Poisson distribution, Poisson (λ). The Poisson distribution defines the probability of the number of events occurring within a certain space (e.g., time or area), where λ is the average number of events per time (or area) unit [8]. The situation is complicated, however, when measuring substance use within the general population, as the majority of individuals likely do not use the substance of interest [6,9]. As such, there is a preponderance of zeros in the data, representing an absence of the behavior of interest. The preponderance of zeros can increase the ratio of variance to mean, an index of dispersion, resulting in overdispersion [10,11]. This is problematic, as the Poisson distribution assumes that the mean and variance are equal (λ). As such, when the count data include a large number of zeros, the distribution may be better defined by a zero-inflated Poisson (ZIP) distribution (Equation (1)) [12–14]. Note that if K (the count of events) is greater than zero in Equation (1), the ZIP distribution, ZIP (π, λ), is Poisson (λ), where π is the probability of structural zeros (known absence of the behavior) and λ is the rate parameter (expected number of events per unit time) [15].

$$pr(X = k) = \begin{cases} \pi + (1 - \pi)e^{-\lambda}, for\ k = 0 \\ (1 - \pi)\frac{\lambda^k e^{-\lambda}}{k!}, for\ k = (1, 2, 3, \ldots). \end{cases} \tag{1}$$

To account for data with a ZIP distribution, researchers conducting longitudinal data analysis (repeated measures data analysis) can model change in two parts, one for nonengagement in the behavior (binary, yes or no), and a second for use (count times engaging in the behavior). Structural equation modeling (SEM) provides modeling options for ZIP data [16]. Essentially, the modeling divides the sample into two parts, with those who do not engage in the behavior and those who progress to engagement, calculating the probability of engagement over time as well as progression in use [17]. One interesting application of this in SEM is two-part modeling [18–20]. An alternative to the LGCM tested here is to consolidate the repeated measures data into a single variable by calculating the area under the curve (AUC) defined by the repeated measures.

Individuals taking advanced mathematics are familiar with the AUC. In calculus, the AUC is the outcome of integration over some range of values, or a summation of polygons within consecutive intervals (e.g., trapezoid rule). Although used frequently in research dealing with biological markers such as daily cortisol [21,22] and receiver operating characteristic (ROC) curves to identify cut points [23–25], the AUC has only recently been employed in other research areas (e.g., dentistry and sleep research) [26,27]. Two equations were proposed for modeling AUC in longitudinal data analysis [28]. The first (Equation (2)) calculates the area under the curve with respect to the ground (AUC—g). This measure calculates the area between the curve (trajectory for longitudinal data) and the x-axis (absence of behavior). Note that the subscripts for the y and x variables represent time points, with 1 representing baseline. The second equation (Equation (3)) calculates the area under the curve with respect to baseline. This is also known as the AUC with respect to the increase (AUC—i). Note that the initial y (outcome; $y_1$) multiplied by the sum of the time intervals is subtracted from the AUC equation to account for the baseline level. Note

also that, for equal time intervals, the summation is the number of time points. This second AUC equation can permit scores below zero, as individuals may use less than they did at baseline. As such, it may be a better way to assess behavioral change even if it is not an area per se.

$$AUC_{Ground} = \left[ \frac{(y_2 + y_1)}{2} \times (x_2 - x_1) \right] + \left[ \frac{(y_3 + y_2)}{2} \times (x_3 - x_2) \right]. \quad (2)$$

$$AUC_{Increase} = \left\{ \left[ \frac{(y_2 + y_1)}{2} \times (x_2 - x_1) \right] + \left[ \frac{(y_3 + y_2)}{2} \times (x_3 - x_2) \right] \right\} - y_1 \sum_{i=1}^{3} t_i, \quad (3)$$

where $t_i$ is the $i^{th}$ interval between consecutive time points.

The purpose of the present study is to assess the performance of the AUC—g and AUC—i models in a prototypical repeated measures analysis design involving predictors of change in substance use across time. We discuss differences in results, comparing our AUC models to a ZIP latent growth curve model with Monte Carlo-simulated data.

## 2. Methods

### 2.1. ZIP LGCM

We generated four separate Monte Carlo models with sample sizes of 500, 250, 100, and 50 and 5000 replications each. M*plus* statistical software (statmodel.com) provides users with examples of different modeling methods in its user's guide [29], including LGCM with ZIP variables. In addition to the examples, M*plus* includes Monte Carlo counterparts to each example. The example chosen was 6.7 (Chapter 6), as it provided the framework for modeling the data of interest—repeated measures of substance use counts. This model was adapted, however, to include two predictor variables, each with a mean of zero and a standard deviation of 1.0. As our ultimate aim was to compare AUC models to an LGCM defined by repeated measures of ZIP random variables, we assessed the simulated variables to ensure they followed the desired ZIP distribution.

LGCM is a latent (unobserved) variable modeling procedure used to assess trajectories across repeated measures of some variable [30]. Besides assessing the nature of trajectories, most researchers are interested in the effects of theoretically relevant predictor variables on markers of change. In LGCM, one models the effect of putative predictor variables on continuous latent variables representing the baseline level (i.e., intercept) and trend (e.g., linear or quadratic) representing repeated measures variables [30]. In this way, one can control for the impact of the predictor variables on the initial level and assess whether they impact the rate of change from baseline.

Unlike LGCM with normal or even categorical random variables, in the case of ZIP data, there are two parts in the LGCM analysis, one part modeling no use versus use (1 = no use, 0 = use) with categorical latent variables and the other part modeling use (number of instances; count data) with continuous latent variables. We based our LGCM on four repeated measures, as using four time points is common in the assessment of adolescent substance use, with at least one measurement point per year of high school, and our primary population of interest is adolescents.

Regarding the population model for our Monte Carlo simulation, we used the following population parameter values. For the count part of the model (those using a substance), the population intercept mean was set at 0.2 with a variance of 0.4, indicating low initial use. With respect to growth, we set the linear trend factor to 0.05, indicating a 0.05 increase in use for each year increase in high school. The trend factor variance was constrained to equal zero. With respect to the binary inflation part of the model (0 = used, 1 = did not use), the intercept factor mean was constrained to zero, with the variance equal to 1. For the inflation slope, the mean was −0.2, with the variance constrained to 0. Thus, we modeled a decreased likelihood of not using over time, with slow growth among users.

With respect to the two putative predictor variables, we used positive and negative values; as in real-world situations, we could have both positive and negative effects on

the intercept and trend. We began with a set of parameter estimates that resulted in high power ($\geq$0.80) in the n = 500 scenario to see how the model would perform with lower sample sizes. We chose 0.80 for power as that is a standard value when calculating sample sizes for grant applications in the health sciences [31]. We set the effect of the first and second variables to 1.0 and $-1.0$ on the count intercept and slope, respectively. For the inflation model part, the effect of the first and second variables on the intercept factor was set to 0.5 and $-0.5$, respectively. For the trend factor, we set the population parameter value to 0.35 and $-0.35$ for the first and second variables, respectively. We saved the data generated with the Monte Carlo simulation for use in assessing the efficacy of the AUC linear regression model.

*2.2. AUC*

In the next stage, we used the data generated in our LGCM simulations to compute the two AUC variables (AUC—i and AUC—g). We then used these variables as dependent variables in linear regression models in M*plus*, with our two Monte Carlo-generated predictor variables as independent variables in our regression models. We used M*plus* to run our multiple regression analysis models because M*plus* permits saving parameter estimates from all analyses for Monte Carlo simulations. This permits researchers to assess bias, coverage, and power for their analyses [See 29]. As such, we ran Monte Carlo simulations using the parameter estimates from our regression analysis models to assess the performance of our AUC regression models with varying sample sizes, focusing on bias, coverage, and power.

The data generated in the ZIP LGCM Monte Carlo simulation were also exported to SPSS statistical software to explore the data using basic descriptive statistics and charts (e.g., histograms). To assess the results of our simulation, we also calculated ZIP parameters lambda (Equation (4)) and pi (Equation (5)) using the method of moments estimation (MME) [12].

$$\lambda_{MME} = \bar{x} + \frac{s^2}{\bar{x}} - 1. \tag{4}$$

$$\pi_{MME} = \frac{(s^2 - \bar{x})}{\bar{x}^2 + (s^2 - \bar{x})}. \tag{5}$$

**3. Results**

Descriptive statistics for each of the four Monte Carlo-generated variables for the different sample sizes appear in Table 1. The table is partitioned into four parts, one part for each sample size (i.e., 500, 250, 100, and 50), and six columns. Within each part are four rows, one for each time point. The six columns provide information on the sample size, the time point, the mean, the standard deviation, and the $\lambda$ and $\pi$ from Equations (4) and (5), respectively. The results suggest that the average use increased over time. The rate ($\lambda$) was highest in the third or fourth time point, depending upon the sample size, with the probability of structural zeros ($\pi$) remaining relatively constant across time and sample size.

Table 2 presents the results of the Monte Carlo analyses of the ZIP LGCMs for the different sample sizes, using 5000 replications, as presented in the M*plus* Monte Carlo output [29]. The four columns represent the latent variables; they are endogenous variables in the LGCM. The rows represent summary values from the four different simulations (one simulation each for an n of 500, 250, 100, and 50). The first and second rows within each simulation present the average of the estimates from the 5000 replications and the percent bias, respectively. The third row is the mean square error (MSE). The last two rows contain the 95% coverage and power values. For the 95% coverage, by the empirical rule, we expect that 95% of all estimates fall within $\pm$1.96 standard deviations of the mean [32].

**Table 1.** Descriptive statistics and ZIP parameters for all four waves for the different sample sizes *.

| N | Wave | Minimum | Maximum | Mean | SD | λ | π |
|---|---|---|---|---|---|---|---|
| 500 | Time 1 | 0 | 83 | 1.93 | 7.419 | 29.45 | 0.93 |
| | Time 2 | 0 | 107 | 2.06 | 8.064 | 32.78 | 0.94 |
| | Time 3 | 0 | 475 | 2.54 | 21.662 | 186.28 | 0.99 |
| | Time 4 | 0 | 206 | 2.57 | 14.417 | 82.45 | 0.97 |
| 250 | Time 1 | 0 | 83 | 2.17 | 8.341 | 33.23 | 0.93 |
| | Time 2 | 0 | 106 | 2.31 | 8.636 | 33.60 | 0.93 |
| | Time 3 | 0 | 38 | 1.73 | 5.153 | 16.08 | 0.89 |
| | Time 4 | 0 | 206 | 3.26 | 18.248 | 104.71 | 0.97 |
| 100 | Time 1 | 0 | 63 | 2.15 | 8.964 | 38.524 | 0.94 |
| | Time 2 | 0 | 43 | 2.04 | 5.605 | 16.440 | 0.88 |
| | Time 3 | 0 | 38 | 2.00 | 5.944 | 18.666 | 0.89 |
| | Time 4 | 0 | 206 | 4.27 | 22.431 | 121.104 | 0.96 |
| 50 | Time 1 | 0 | 63 | 2.14 | 9.165 | 40.391 | 0.95 |
| | Time 2 | 0 | 43 | 2.60 | 7.100 | 20.989 | 0.95 |
| | Time 3 | 0 | 32 | 1.36 | 4.681 | 16.472 | 0.92 |
| | Time 4 | 0 | 206 | 6.90 | 31.403 | 148.820 | 0.95 |

* Means and variances are used to calculate the ZIP parameters π and λ.

**Table 2.** Monte Carlo results of LGCM.

| n | Measures [5] | Count Intercept [1] | | Binary Intercept [2] | | Count Trend [3] | | Binary Trend [4] | |
|---|---|---|---|---|---|---|---|---|---|
| | | $X_1$ | $X_2$ | $X_1$ | $X_2$ | $X_1$ | $X_2$ | $X_1$ | $X_2$ |
| 500 | Average | 1.0009 | −0.9993 | 0.5091 | −0.509 | 0.0998 | −0.1002 | 0.3545 | −0.3537 |
| | %Bias | 0.09 | −0.07 | 1.82 | 0.018 | −0.2 | 0.2 | 1.2857 | 1.0571 |
| | MSE | 0.0032 | 0.0032 | 0.0331 | 0.331 | 0.0005 | 0.0005 | 0.106 | 0.011 |
| | Coverage | 0.942 | 0.943 | 0.95 | 0.946 | 0.93 | 0.931 | 0.952 | 0.943 |
| | Power | 1.00 | 1.00 | 0.829 | 0.846 | 0.992 | 0.992 | 0.939 | 0.933 |
| 250 | Average | 1.0022 | −0.9996 | 0.5257 | −0.5203 | 0.1001 | −0.1007 | 0.3595 | −0.3609 |
| | %Bias | 0.22 | −0.04 | 5.14 | 4.06 | 0.1 | 0.7 | 2.7143 | 3.1143 |
| | MSE | 0.0066 | 0.0067 | 0.0738 | 0.0732 | 0.001 | 0.0011 | 0.0229 | 0.0237 |
| | Coverage | 0.934 | 0.939 | 0.943 | 0.949 | 0.924 | 0.925 | 0.948 | 0.945 |
| | Power | 1.0 | 1.0 | 0.524 | 0.514 | 0.882 | 0.888 | 0.691 | 0.692 |
| 100 | Average | 1.0039 | −1.004 | 0.5705 | −0.5599 | 0.1013 | −0.1015 | 0.3804 | −0.3846 |
| | %Bias | 0.39 | 0.4 | 14.1 | 11.98 | 1.3 | 1.5 | 8.6857 | 9.8857 |
| | MSE | 0.0193 | 0.0187 | 0.2371 | 0.2274 | 0.0034 | 0.0032 | 0.0735 | 0.0737 |
| | Coverage | 0.922 | 0.923 | 0.942 | 0.95 | 0.901 | 0.905 | 0.936 | 0.943 |
| | Power | 1 | 1 | 0.203 | 0.189 | 0.543 | 0.543 | 0.347 | 0.348 |
| 50 | Average | 1.0131 | −1.0123 | 5.2436 | −3.6967 | 0.1007 | −0.102 | −0.1756 | −0.18 |
| | %Bias | 1.31 | 1.23 | 948.72 | 639.34 | 0.7 | 2.00 | −150.171 | −48.571 |
| | MSE | 0.0494 | 0.0473 | 25461.42 | 7943.048 | 0.0093 | 0.0092 | 1909.731 | 409.3025 |
| | Coverage | 0.893 | 0.9 | 0.935 | 0.937 | 0.886 | 0.886 | 0.923 | 0.921 |
| | Power | 0.986 | 0.985 | 0.112 | 0.111 | 0.334 | 0.343 | 0.205 | 0.204 |

[1] Intercept factor for count part; [2] intercept factor for the binary part; [3] linear trend factor for the count part; [4] linear trend factor for the binary part; [5] $X_1$ and $X_2$ are the two continuous predictor variables.

Percent bias values are lower for the parameter estimates in the count model part than the binary use model part for all sample sizes. As the sample size decreases, the percent bias increases, peaking with n = 50, especially for the binary model part with bias as high as 949%. MSE is higher for the binary than the count model parts, increasing inversely with sample size, peaking at 7943.048 with n = 50 for the effect of variable $X_2$ on the intercept. Power decreased particularly for the binary model part and for sample sizes 100 and below, with power values no higher than 0.348 for the binary part growth factor (effect of $X_2$ on slope; n = 100) and 0.205 for the binary part growth factor (effect of $X_1$ on slope; n = 50).

## 3.1. Area under the Curve

Table 3 presents descriptive statistics for the two AUC measures, along with the natural log-transformed AUC—g measure. Means, standard deviations, skewness, kurtosis, the median, and the interquartile range (IQR) are also shown. These values suggest that both measures are not normally distributed. To assess this possibility, we also ran Kolmogorov–Smirnov (Lilliefors significance correction) and Shapiro–Wilk tests of normality. For all four sample sizes, the results suggest non-normal data ($p < 0.001$). Even with the natural log-transformed AUC—g variable, the results suggest non-normality, $p < 0.001$.

**Table 3.** Descriptive statistics for the AUC variables.

| n | AUC Measure | Mean | SD | Skewness | Kurtosis | Median | IQR [1] |
|---|---|---|---|---|---|---|---|
| | AUC—i | 1.061 | 25.589 | 12.146 | 240.570 | 0.00 | 2.00 |
| 500 | AUC—g | 6.857 | 26.907 | 12.565 | 196.822 | 1.50 | 4.50 |
| | LN AUC—g | 1.11 | 1.13 | 1.089 | 1.067 | 0.916 | |
| | AUC—i | 0.24 | 16.343 | −3.515 | 45.006 | 0.00 | 2.50 |
| 250 | AUC—g | 6.756 | 20.107 | 7.818 | 77.213 | 1.50 | 5.00 |
| | LN AUC—g | 1.158 | 1.135 | 0.991 | 0.709 | 0.916 | 1.79 |
| | AUC—i | 0.8 | 20.416 | −3.714 | 38.084 | 0.00 | 2.50 |
| 100 | AUC—g | 7.25 | 18.484 | 4.754 | 26.398 | 1.50 | 4.50 |
| | LN AUC—g | 1.164 | 1.181 | 1.087 | 0.708 | 0.916 | 1.70 |
| | AUC—i | 2.06 | 15.441 | 22.673 | 8.48 | 0.00 | 4.13 |
| 50 | AUC—g | 8.48 | 23.780 | 18.467 | 1.105 | 1.250 | 4.13 |
| | LN AUC—g | 1.105 | 1.251 | 1.340 | 1.535 | 0.805 | 1.63 |

[1] IQR; interquartile range.

Table 4 presents the results of the linear Monte Carlo estimations run with 500, 250, 100, and 50 participants with natural log-transformed AUC—g and AUC—i as the dependent variables. We natural log-transformed AUC—g as an analysis of the descriptive statistics indicated divergence from normality. By contrast, we did not transform AUC—i because it contained negative values, precluding taking its natural log. For AUC—g, bias and the MSE remain low for all models, with bias never exceeding 0.41% and the MSE never exceeding 0.02. As expected, and by design, power is 1 for the n = 500 model but never decreases below 0.80 (n = 50 model). Coverage never decreased below 0.93 regardless of sample size.

With AUC—i as the dependent variable, for the largest sample size (n = 500), bias did not exceed 2%. However, there was much instability in parameter estimation in this model. Indeed, as the sample size decreased, bias varied, being highest for n = 250 and n = 50 and reaching a high of 71% (n = 250). MSE increased inversely with sample size, particularly for n = 100 and n = 50, reaching a high of 4.76 with n = 50. Power was low throughout, never exceeding 0.31. Coverage was adequate and never decreased below 0.928.

## 3.2. Case Examples

To better understand the difference between analyzing our data with the area under the curve and latent growth curve modeling, we selected two cases and plotted their values based on the AUC measures and LGCM (Figures 1–4). Figure 1 presents the estimated and observed values for substance use based on the ZIP LGCM. Although the lines are clearly different, the estimated trajectory shows growth from time 1 to time 4, akin to the observed change. Figure 2 presents the concomitant AUC histograms for AUC—i, AUC—g, and LN AUC—g. There is a distinct difference between panels B and C, with greater variability evident in panel C than B, suggesting that the natural log-transformed AUC—g performed better than the raw AUC—g score.

**Table 4.** Monte Carlo simulation regression results for AUC—g and AUC—i.

| n | Measures | AUC—g [1] | | AUC—i | |
|---|---|---|---|---|---|
| | | $X_1$ | $X_2$ | $X_1$ | $X_2$ |
| 500 | Average | 0.3282 | −0.4147 | 0.9069 | −1.4914 |
| | Bias | −0.243 | −0.072 | −2.274 | −0.441 |
| | MSE | 0.0021 | 0.0018 | 1.3694 | 1.1977 |
| | Coverage | 0.945 | 0.95 | 0.946 | 0.95 |
| | Power | 1 | 1 | 0.134 | 0.283 |
| 250 | Average | 0.2924 | −0.4504 | −0.5813 | 0.0029 |
| | Bias | −0.205 | 0.0889 | 1.6259 | −71 |
| | MSE | 0.0045 | 0.004 | 1.1947 | 1.0654 |
| | Coverage | 0.947 | 0.948 | 0.947 | 0.948 |
| | Power | 0.991 | 1 | 0.091 | 0.051 |
| 100 | Average | 0.3167 | −0.4294 | −2.9642 | −2.0807 |
| | Bias | −0.409 | −0.14 | 0.8918 | −0.54 |
| | MSE | 0.0115 | 0.01 | 4.448 | 3.8672 |
| | Coverage | 0.941 | 0.944 | 0.941 | 0.944 |
| | Power | 0.845 | 0.986 | 0.313 | 0.195 |
| 50 | Average | 0.4064 | −0.4795 | 0.1414 | −1.385 |
| | Bias | −0.392 | 0.1044 | −14.303 | 0.581 |
| | MSE | 0.0226 | 0.021 | 4.7604 | 4.4243 |
| | Coverage | 0.936 | 0.929 | 0.936 | 0.928 |
| | Power | 0.802 | 0.924 | 0.069 | 0.125 |

[1] Natural log-transformed.

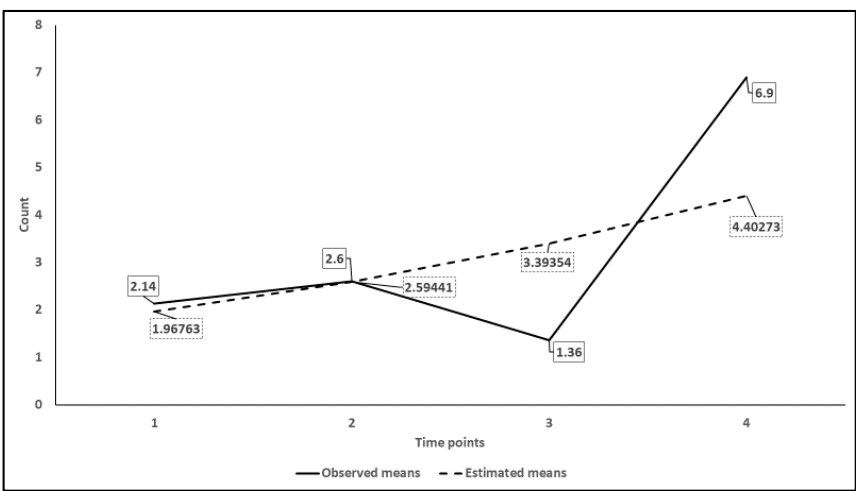

**Figure 1.** Observed and estimated trajectories from the ZIP LGCM for the N = 50 simulation.

Figure 3A presents the observed counts across the four time points for simulated participant 10. Figure 3B presents the concomitant AUC figures (bar charts for AUC—i, AUC—g, and LN (AUC—g)). Figure 4B presents the same charts for simulated participant 18. A notable difference is seen in the AUC—i bar in Figure 3B. Given the decline in use from time 2 onward and the participant having zero use in the final two time points, the AUC—i value is negative (see Equation (3) to understand the calculation of this AUC measure). The same is not evident for simulated participant 118 (Figure 4). Although there is also a large decline in use, there was an increase in use from baseline to the third time point. Further, given that there is only one time point with no substance use (time point 4) and the nature of the equation for calculating this AUC measure (Equation (3)), this value is positive.

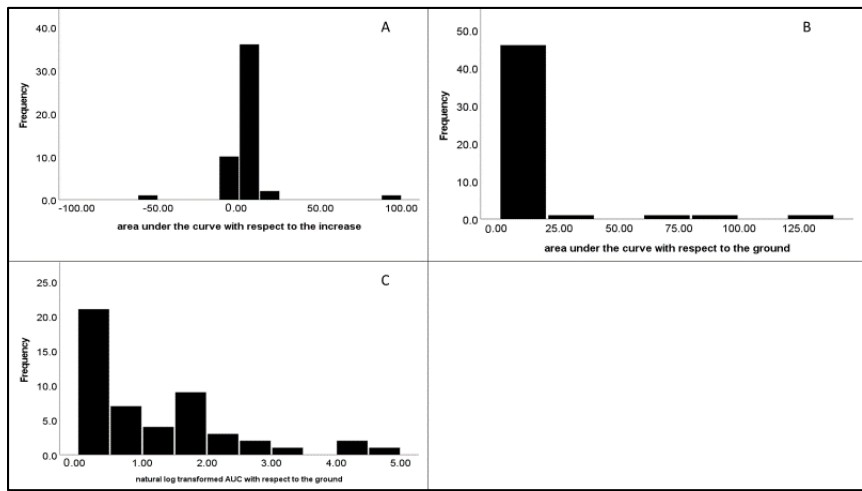

**Figure 2.** Histograms for the three area under the curve (AUC) measures for the N = 50 simulation. (Panel **A**): AUC—i; (Panel **B**): AUC—g; (Panel **C**): LN AUC—g.

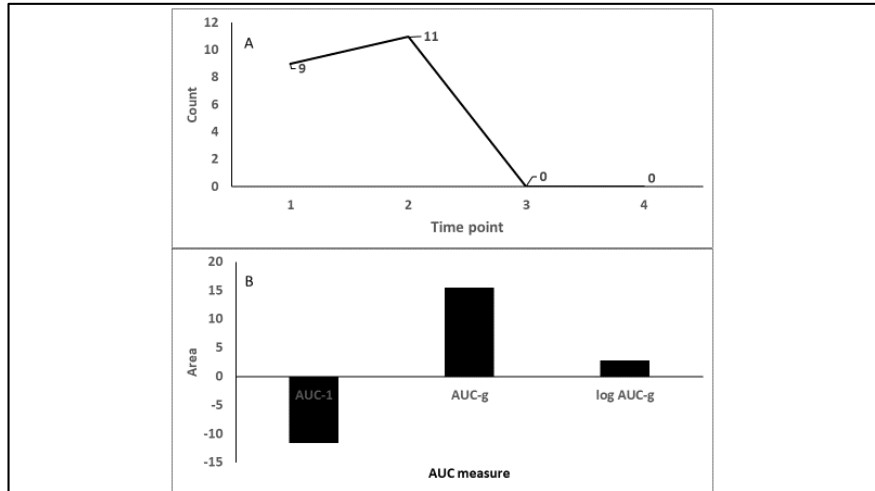

**Figure 3.** Observed values across three time points (observed trajectories; Panel **A**) and calculated AUC values (Panel **B**) for individual 10 from the N = 50 simulated data.

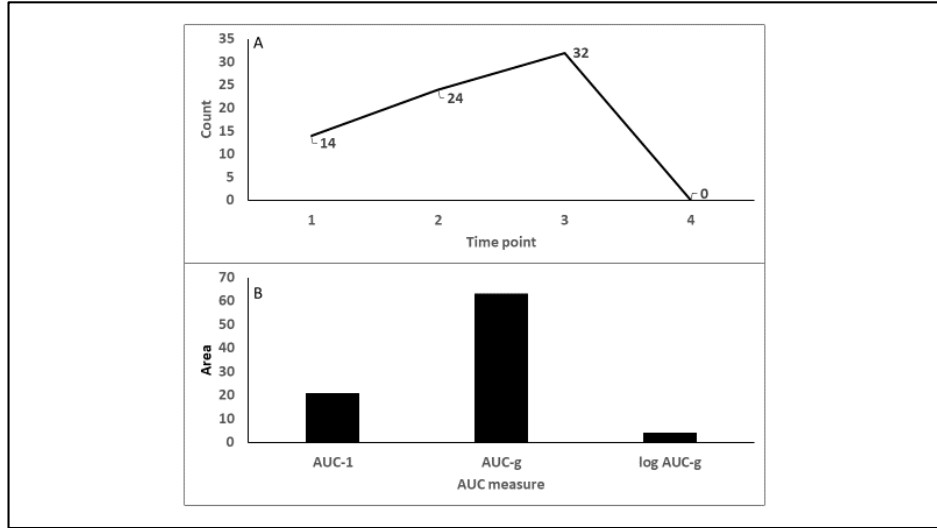

**Figure 4.** Observed values across three time points (observed trajectories; Panel **A**) and calculated AUC values (Panel **B**) for individual 18 from the N = 50 simulated data.

## 4. Discussions

The results of our simulation study suggest that the area under the curve (AUC) may be a viable alternative for researchers dealing with count data with a preponderance of zeros, even when the sample size is limited. However, our results suggest that the AUC's efficacy depends upon which method is employed in its calculation. We used two equations to calculate the AUC (Equations (2) and (3)). Equation (2) calculates the AUC with respect to the ground. It is the more traditional calculation of the AUC. Equation (3) calculates the area with respect to the baseline. As the product of the baseline level and time is subtracted from the AUC calculation, this measure is better interpreted as an indicator of change than area. Indeed, negative values are possible, as values can decrease below the baseline level, particularly if one reduces or quits a substance (e.g., quits or cuts down vaping). As such, further studies are necessary to validate this AUC measure as an efficacious indicator of change.

A key finding of this study is that bias was low and power was high when using AUC—g, even with small samples (n = 50). Although the nature of the parameters differs when comparing LGCM and AUC—g, it is notable that the ZIP LGCM simulation with the smallest sample size resulted in high bias and low power estimates, meaning that this method may be less than ideal when dealing with zero-inflated data and a small sample size. This is critical when researchers lack the resources needed to collect large samples. Further, the use of methods such as LGCM requires specialized software, which can be expensive or difficult to code when free. By contrast, AUC calculations are simple to program into existing software and require simple coding.

### 4.1. Limitations and Conclusions

The results of this study must be viewed with respect to its limitations. First, we only assessed ZIP random variables. Although common in substance use research, the performance of the two AUC variables may differ with other types of non-normal random variables encountered in substance use research. Second, the simulated data were generated using a ZIP latent growth curve model (LGCM) and then employed in a regression analysis to generate parameter estimates for the AUC simulations. Using other methods to simulate the data may lead to different results. Third, different population parameters for the initial simulations may generate different patterns of simulated substance use variables.

With these limitations in mind, the results of this study provide some preliminary data regarding the efficacy of the AUC when dealing with zero-inflated Poisson (ZIP) data. Future research must compare AUC models to other analysis strategies when dealing with repeated measures of substance use count data beyond LGCM, including GEE. Further, researchers must assess the AUC for other variable types beyond ZIP data. In addition, researchers should assess the distributional properties of the two AUC variables, particularly when dealing with diverse types of repeated measures variables (e.g., ZIP versus exponential), and explore other equations that may perform better particularly when accounting for declining use instead of merely subtracting baseline use as in the AUC-i equation.

### 4.2. Software

Mplus software is not freely available, although one can download a demo version that permits six dependent variables and two independent variables, (https://www.statmodel.com/demo.shtml (accessed on 16 February 2023).

**Funding:** This research received no external funding.

**Institutional Review Board Statement:** This study used simulated data only. Therefore, it did not require Institutional Review Board approval.

**Informed Consent Statement:** Not applicable.

**Data Availability Statement:** Data are available from author upon email request.

**Conflicts of Interest:** The author declares no conflict of interest.

## Abbreviations

| Abbreviation | Definition |
| --- | --- |
| GEE | General Estimating Equation |
| LGCM | Latent Growth Curve Model |
| ZIP | Zero-Inflated Poisson |
| SEM | Structural Equation Modeling |
| AUC | Area Under the Curve |
| AUC—g | Area Under the Curve with respect to ground |
| AUC—i | Area Under the Curve with respect to the increase |
| ROC | Receiver Operating Characteristic |

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
