# Peer review of "Assessing Area under the Curve as an Alternative to Latent Growth Curve Modeling for Repeated Measures Zero-Inflated Poisson Data: A Simulation Study"

_stats, doi:10.3390/stats6010022_

Round 1
Reviewer 1 Report (Previous Reviewer 2)
The 4 figures should be made more readable by increasing the size of the symbols within the figure, in the legend and on the axis labels
Author Response
Reviewer comment: "The 4 figures should be made more readable by increasing the size of the symbols within the figure, in the legend and on the axis labels"
Author's response: Thank you for the comments. Looking at the figures again, I realize that the text is very small, making it difficult to read. I therefore changed the font to 16. I also made sure the labels (numbers) within the figures are better - placed to improve clarity. See figures within the revision for the changes
Reviewer 2 Report (New Reviewer)
Please see the PDF file attached for my comments.

Author Response
Reviewer Comments: "My main concern lies in the introduction section where I believe prevalent existing statistical techniques deserve more discussion and elaboration. Indeed, this paper primarily presented a simulation study. Still, typical methods that deal with longitudinal data such as generalized estimating equations and linear mixed effects models should be elaborated to improve
readership. More detailed comments are as follows."
Author response: I agree. I only provided a brief discussion of these methods in preparation for the presentation of the simulation study. I therefore increased the depth of discussion of these typical methods in the introduction.
1. The first paragraph of the introduction needs some restructuring and/or elaboration. At the very beginning of the introduction section, the author shall state “clustered data” or “longitudinal data” explicitly to make readers clear about the type of analysis the paper is going to discuss. More importantly, current popular approaches such as generalized estimating equations and linear mixed effect models deserve more discussion and citation.
Proposed by Liang and Zeger [1], generalized estimating equations can uncover the effect of interest from a population-average perspective when analyzing longitudinal data.
Moreover, they can incorporate missing data for analysis and design, as shown in Robins
et. al. [3] and Yang et. al. [4]. On the other hand, linear mixed effect models are good
at characterizing individual variability by subject-specific random effects and have been
applied in numerous studies, including Morrell et. al. [2]. These papers should be cited
to provide sufficient background of the existing longitudinal methods before introducing
the proposed zero-inflated Poisson scenario.
2. If publicly available or available upon request, the author shall state availability of the
program or software used in the paper at the end of the manuscript.
Minor points:
1. Line 8: It should be “generalized estimating equations” instead of “general estimating
equations”.
1
2. Keyword “Clustered data” or “Longitudinal data” should be added.
3. Line 177: Greek letters λ and π should be used.
4. Line 190-191: I don’t think the definition of power is needed for a statistical journal.
5. Line 195: Mean squared error is defined twice (Acronym MSE has already appeared in
Line 188).
6. Lines 212: Sentence “We did not natural ...” should be edited for grammar.
7. Figure 1 and Figure 2: These two figures can be bigger and centered, as Figure 3 and
Figure 4.
Despite the concerns described above, I would suggest considering the suggestions to improve
the manuscript for publication.
Author response: Please see the attached document for my response to your very helpful comments.

Round 2
Reviewer 2 Report (New Reviewer)
The author has been responsive to earlier comments, and I quite like the revised paper.
This manuscript is a resubmission of an earlier submission. The following is a list of the peer review reports and author responses from that submission.
Round 1
Reviewer 1 Report
The authors introduced a new paper which has title (Assessing Area Under the Curve as an alternative to latent growth curve modeling for repeated measures zero inflated Poisson data: A simulation study)
The paper can be accepted after the following major comments:-
1- All first letter in the title should be in capital letter.
2-Abstact should not contains abbreviations
3- Make table of abbreviations in the end of the paper.
4- All equations must be numbered numerically by the standard way
5-all plots not clear please improve it
6- the second half of the first page is empty WHY?
7- The paper need to improve the language many error on it
8- All equations should be ended by qom or dot.
9- All tables need to improvements
10- conclusion section is absent
Reviewer 2 Report
Report on stats-1996013
Title: Assessing Area Under the Curve as an alternative to latent growth curve modeling for repeated measures zero inflated Poisson data: A simulation study
1. Brief summary of the content of the manuscript
In this paper, the author focuses on comparing the relative efficacy of the two area under the curve (AUC) measures to latent growth curve modeling (LGCM) for the assessment of substance use counts. This is done using simulated data for repeated measures of ZIP variables at four time points. Four models are generated with different sample sizes and 5,000 replications each. The statistical software used is Mplus. According to the author, this software provides users with different modeling methods.
2. Reasoning behind my recommendation
Many points have to be clarified with respect to the distributional properties of the AUC
3. Provide more lists of your minor for the improvement of the manuscript.
- In the introduction, the author says “The Poisson is ideal for count data” . This sentence has to be modified and the author should talk about under and overdispersion count distribution with some references.
- In equation 1, p is not equal to P(X=0) :
- In equation 1, p is the probability of structural zeros. The ZIP distribution is a mixture between a point mass at zero and a Poisson distribution.
- Page 3 : Providing an internet address for the Mplus software would be a great utility.
- Page 4 : In order to verify the veracity of the simulations (i.e., the generation of ZIP distributed variables), the sample estimates given by equations 4 and 5 do not seem enough, compared to dispersion indicators.
- The figures 1 to 4 are illisible.
- The author should explain why he does not take into account the expected value of the AUC under known models.
Reviewer 3 Report
This paper advocates for using an area under the curve (AUC) approach to modeling substance use trajectories where the outcome variable is distributed as a zero inflated poisson (ZIP) random variable. The scientific argument to consider AUC as an alternative to latent growth curve modeling is interesting, however, the presented statistical argument is lacking in several ways.
Specifically, I have the following major concerns:
1) Although the introduction is reasonably well justified in terms of why researchers should consider AUC for modeling longitudinal count data for substance use trajectories, there was no motivating example data and the author simply used an example dataset from Mplus defaults to generate longitudinal ZIP data. Without motivation or a clearer understanding of the defaults Mplus is using, it is very hard to follow the type of data being generated in this example.
2) The author seems to compare the power between predictors in a latent growth curve model and predictors in a model with AUC as the outcome and conclude that the AUC approach has “better” power, particularly for small sample sizes. However, these two models are estimating dramatically different parameters with very different scientific interpretations, so it makes little sense to compare the two of them on a statistical consideration such as power. The author would be better served to illustrate the similarities and differences in interpretation of these parameters in both models (on the same dataset) and encourage researchers to carefully consider which interpretation makes more scientific sense.
3) It is not clear to me how or why the “population” effects for the AUC values (in tables 3 and 4) are changing with the sample size? Presumably these values are the same in truth for all sample sizes if they are population parameters? This confusion is most notable with the results for AUC-i (table 4) where the “population” effects for X1 and X2 are dramatically different with the different sample sizes.
4) Given the concerns above, the discussion is overstated in terms of statistical importance and justification for using AUC. I think it is an interesting and worthwhile argument for researchers to consider AUC as an approach to these types of data; I just don’t think “better power” (as illustrated here) is a useful justification when it estimates something so different from an LGCM.
Minor comments:
1) The abstract asserts that substance use counts “follow a zero inflated Poisson distribution” which is almost certainly not true in every instance. Suggest editing to something like “may approximate a zero inflated Poisson distribution” or otherwise rephrasing to reflect the uncertainty.
2) The tables are exceptionally difficult to read – I would suggest reorganizing the tables to have fewer columns per table and to allow for easy vertical (within row) coomparisons of the most relevant information.
3) It is not clear what information the reader is supposed to take from Figure 1-4? Looking actual values from the same individual over time (particularly for n=50) would be far more informative and would help illustrate the difference between both AUC approaches and the LGCM approach.
4) The methods and results sections are not well defined – a lot of detail of the simulations and how the simulations were evaluated is included only in the results section and should be in the methods instead.
Round 2
Reviewer 1 Report
The paper can be publish in the current form.
Reviewer 2 Report
line 50 : replace "average number of events" with "average number of events per time (or area) unit"
line 63 : replace "average events in a unit time" with "expected number of events per time unit"